# The Role of Calcium and Iron Homeostasis in Parkinson’s Disease

**DOI:** 10.3390/brainsci14010088

**Published:** 2024-01-17

**Authors:** Ji Wang, Jindong Zhao, Kunying Zhao, Shangpeng Wu, Xinglong Chen, Weiyan Hu

**Affiliations:** 1School of Chinese Materia Medica & Yunnan Key Laboratory of Southern Medicine Utilization, Yunnan University of Chinese Medicine, Kunming 650500, China; wj9811288@163.com; 2School of Pharmaceutical Science & Yunnan Key Laboratory of Pharmacology for Natural Products, Kunming Medical University, Kunming 650500, China; 20210205@kmmu.edu.cn (J.Z.); zky2697@163.com (K.Z.); wushangpeng2000@163.com (S.W.)

**Keywords:** Parkinson’s, calcium, ferroptosis, homeostasis

## Abstract

Calcium and iron are essential elements that regulate many important processes of eukaryotic cells. Failure to maintain homeostasis of calcium and iron causes cell dysfunction or even death. PD (Parkinson’s disease) is the second most common neurological disorder in humans, for which there are currently no viable treatment options or effective strategies to cure and delay progression. Pathological hallmarks of PD, such as dopaminergic neuronal death and intracellular α-synuclein deposition, are closely involved in perturbations of iron and calcium homeostasis and accumulation. Here, we summarize the mechanisms by which Ca^2+^ signaling influences or promotes PD progression and the main mechanisms involved in ferroptosis in Parkinson’s disease. Understanding the mechanisms by which calcium and iron imbalances contribute to the progression of this disease is critical to developing effective treatments to combat this devastating neurological disorder.

## 1. Introduction

PD (Parkinson’s disease), the second most common multifactorial progressive neurodegenerative disease after Alzheimer’s disease, attracts widespread attention due to the rising prevalence as the world population ages and lives longer [1,2].

Though the cause of PD is currently unknown, it is now generally accepted that aging is the most influential risk factor and various environmental and genetic factors have also been shown to influence Parkinson’s risk [3,4,5]. However, age and genetic and environmental factors can only explain a limited part of PD [6]. The causes of most cases of PD can be complex and could be the result of the interplay of age, genetic susceptibility, and environmental factors.

Iron and calcium are found in low concentrations in the body, but they play an essential and irreplaceable role. Excessive metal accumulation in the central nervous system may induce oxidative stress, disrupt mitochondrial function, and damage different types of proteins and receptors [7,8]. Various studies have shown that changes in metal ion homeostasis may lead to damage to the CNS (central nervous system) with neurodegeneration, disability, and neuroinflammation, contributing to the development of several neurodegenerative diseases [9]. Growing evidence indicates that calcium overload and iron deposition play a vital role in the pathogenesis of Parkinson’s disease [10,11,12].

In this paper, we focus on the pathophysiological role of calcium and iron ions and their homeostatic imbalance in pathogenesis, exploring the therapeutic potential of regulating calcium and iron homeostasis in Parkinson’s disease.

## 2. Calcium and PD 

Ca^2+^ is the second messenger of all living things [13]. When the cells are in a resting state, the calcium ion concentration in the cytoplasm is 100 nmol·L^−1^, the calcium ion concentration in the endoplasmic reticulum is about 0.5–1 mmol·L^−1^, and the extracellular calcium ion concentration is 1–2 mmol·L^−1^. The concentration of Ca^2+^ ions in the cytoplasm is 20,000 times lower than the concentration in the extracellular space [14]. This gradient allows cells to use Ca^2+^ as an effective intracellular signal to respond to and adapt to rapidly changing extracellular and intracellular environments. Cells can alter various physiological functions by activating or inhibiting Ca^2+^-dependent signal transduction pathways. In neurons, Ca^2+^ motion can occur across the plasma membrane, either through electrical activity or through agonists like the DHP (dihydropyridine channel) derivative BAY K8644, which increases calcium influx through L-type calcium channels [15]. Ca^2+^ signaling affects all aspects of neuronal cell biology, so calcium homeostasis must be tightly controlled [16,17].

Intracellular Ca^2+^ levels, Ca^2+^ microdomains, Ca^2+^ buffering, and Ca^2+^ entry patterns are tightly controlled in SN (Substantia nigra) DA(dopaminergic) neurons because Ca^2+^ regulates multiple cellular functions such as excitability, neurotransmitter release, ATP production [18], apoptosis, and the general regulation of enzymes and gene expression [19]. Defects in calcium processing play an essential role in aging and neurodegeneration, with neurons relying on calcium ions to regulate key processes such as neurogenesis, neurotransmission, synaptic plasticity, and gene transcription [16]. In the investigation of individuals with PD and healthy controls, it was observed that PD patients exhibit a higher average calcium level compared to their healthy counterparts. This elevated calcium level has been associated with an increased susceptibility to Parkinson’s disease due to its role in stimulating excessive dopamine biosynthesis, thereby leading to dopaminergic neurotoxicity [20].

Ca^2+^ is regulated by various pathways [21]. *α*-synaptic nucleoprotein aggregation, an important pathologic feature of Parkinson’s disease, leads to disruption of calcium homeostasis [17]. The Ca^2+^ of SN DA neurons can pass through channels in the plasma membrane (voltage-gated calcium channels, VGCC) and transporters (glutamate receptors, NMDA (N-methyl-D-aspartic acid receptor)-R) or AMPA (α-amino-3-hydroxy-5-methyl-4-isoxazole-propionic acid receptor)-R) from extracellular to intracellular [22]. In the cell, Ca^2+^ is also released from mitochondria (mitochondrial calcium transfer protein, MCU) and ER (Endoplasmic reticulum) through RyR (Ryanodine receptor) and IP_3_ (Inositol 1,4,5-trisphosphate) receptors), or through storage-operated calcium channels such as STIM (Stromal interaction molecule)/Orai (Calcium release-activated calcium modulator) and transient receptors potential channel complexes [23,24] (Figure 1).

Ten types of VGCCs are distinguished based on their distinct pharmacological and functional properties, each containing a different pore-forming α1 subunit. The subcellular expression, activity, gating behavior, and pharmacological properties of VGCCs are tuned by extensive alternative splicing and their association with modulatory accessory subunits (β1–β4, α2δ1–α2δ4) [25]. Based on their α1 subunit sequence homology as well as their functional and pharmacological properties, three VGCC families have been defined: Cav1, Cav2, and Cav3. The Cav1 family is composed of four LTCC members (Cav1.1–Cav1.4), which are characterized by their sensitivity to low nanomolar concentrations of DHPs [26]. Cav1.2 and Cav1.3 in the L-type voltage-gated Ca^2+^ channels are associated with PD [27] and Cav1.2 is prevalent in juvenile SNc (Substantia Nigra) DA neurons, but in senescent SNc DA neurons, Cav1.3 is preferentially used for Ca^2+^ inflow, allowing Ca^2+^ to enter through an oscillatory pathway that contributes to the membrane potential threshold, which is the basis of autonomous pacing [28]. Autonomic pacing is a distinctive active state displayed by dopaminergic neurons, characterized by the presence of a spontaneous, slowly depolarizing membrane current that sustains their basal activity level. This mechanism ensures an uninterrupted supply of dopamine to interconnected brain regions, which is crucial for maintaining baseline dopamine levels in the striatum. Unlike Cav1.2, the working range of Cav1.3 does not allow for complete closure of Cav1.3 channels during pacing, which results in elevated intracellular Ca^2+^ levels, and continuous Ca^2+^ influx is essential for regulating physiologic DA release from SNc DA neurons [29]. It is necessary, however, that the chronic and excessive presence of Ca^2+^ may synergize with several risk factors (i.e., aging, mitochondrial toxins, mutations) and produce metabolic stress and mitochondrial damage. The increased expression of Cav1.3 in the cerebral cortex of patients with early Parkinson’s disease occurs before pathological changes, suggesting that calcium homeostasis disorders are not only a compensatory consequence of neurodegenerative processes [30], but also an early feature of Parkinson’s disease. DHP antihypertensive drugs appear to be attractive agents for neuroprotective and preventive treatment of PD [31,32]. Epidemiological studies have shown that administration of DHP-type Ca^2+^ channel blockers reduces the risk of PD in late life by 20 to 30% [33,34]. In contrast, members of the Cav2 family (Cav2.1–Cav2.3), mediating P/Q-, N-, and R-type voltage-gated Ca^2+^ currents, are located presynaptically and are required for fast neurotransmitter release [35]. Low voltage-activated Cav3 channels (Cav3.1–Cav3.3) compose the family of T-type Ca^2+^ channels (TTCCs). TTCCs activate and inactivate at more negative potentials than Cav1 and Cav2 channels. This negative operation range allows them to be active at subthreshold voltages and provides them with a prominent role in the control of neuronal firing patterns [36].

The activity of dopaminergic neurons is regulated by Ca^2+^ carriers and influenced by Ca^2+^ homeostasis [37]. About 80% of calcium is stored in organelles (such as mitochondria, sarcoplasmic reticulum, endoplasmic reticulum, etc.). Mitochondria and the endoplasmic reticulum are the main intracellular organelles that regulate calcium, but evidence suggests that other organelles, such as lysosomes and the Golgi apparatus, also act as important intracellular stores of Ca^2+^ [38]. 

Among them, ER (the endoplasmic reticulum) is considered to be the main reservoir of intracellular Ca^2+^, so the calcium signaling conveyed by ER is crucial in cell conduction. For example, SOCE (store-operated calcium entry) is one of the crucial channels that mediate extracellular Ca^2+^ into the cell, which is due to a large reduction in calcium ions in the lumen of the endoplasmic reticulum, which, in turn, triggers extracellular Ca^2+^ to flow into the cytoplasm and then supplements Ca^2+^ back to ER through the SERCA (ATPase sarco-ER Ca^2+^-ATPase) [39]. The occurrence of this process is mediated by the STIM1 protein in the endoplasmic reticulum matrix and the Orai calcium channel on the plasma membrane. Genetic mutations in Orai1 or STIM1 often lead to SOCE channel defects, which are closely related to the occurrence of various diseases [40,41]. There are two subtypes of STIM proteins, STIM1 and STIM2, whose calcium affinity has been debated in the field for many years. STIM1 is thought to regulate the classical SOCE process [42], while STIM2, due to its lower Ca^2+^ affinity, can sense more subtle changes in Ca^2+^ concentration and play a fine-tuning role [43]. STIM1, a type of transmembrane protein mainly located in the endoplasmic reticulum, is the calcium sensor in the endoplasmic reticulum, mainly promoting Ca^2+^ transport [44]. Significant reductions in STIM1 protein levels have been found in brain tissue in AD (Alzheimer’s disease) patients, while in PD patients, complexes formed by STIM1 and TRPC1 (transient receptor potential channel 1) can inhibit CaV1.3 channels, leading to disruption of neuronal Ca^2+^ homeostasis [45], ultimately leading to the development of PD symptoms. STIM2 is more widely distributed in the hippocampus and is involved in dendrite production [46], maturation, and stabilization through the important calcium/calmodulin-dependent protein kinase or calmodulin kinase pathways [47]. Syt7 (synaptotagmin 7) is a calcium ion detector that dynamically increases neurotransmitter release [48]. It has been demonstrated that STIM2 is mostly present at the presynaptic end, and SOCE can enhance spontaneous neurotransmission at excitatory synapses by facilitating the release of syt7-mediated neurotransmitters through STIM2 [24,49]. Long-term increased self-release can lead to severe synaptic dysfunction and apoptosis [50]. Endoplasmic reticulum stress has been found in a variety of neurodegenerative diseases, and it is likely that the presynaptic SOCE process also secretly contributes its own dark power to these diseases [51,52,53]. This sheds light on the possible impact of a mechanism of neurotransmitter release regulated by endoplasmic reticulum calcium ions on Parkinson’s disease and opens new possibilities for the treatment of neurodegenerative diseases [54].

The Ca^2+^ concentration in mitochondria mainly depends on the ER-mitochondria-associated membrane (MAM) and mitochondrial pathways [55]. Mitochondrial Ca^2+^ levels are strongly correlated with ER, as 20% of mitochondria are closely associated with the endoplasmic reticulum [56]. This region is called MAM, and the two are usually about 10–25 nm apart, which regulates the influx of calcium ions into the mitochondrial matrix [57]. The specific mechanism is through the IP_3_-VDAC1 (voltage-dependent anion channel 1) pathway, calcium ions in the endoplasmic reticulum can be directly released through IP_3_R (IP 3 receptor), enter the cytoplasm from the endoplasmic reticulum, and then enter mitochondria through the calcium ion channel VDAC1 on the outer mitochondrial membrane [58,59]. VDAC1 and IP_3_R control the permeability of the endoplasmic reticulum and mitochondria to calcium ions, respectively, so the abnormal expression of the two will have a great influence on the distribution of calcium ions. Previous studies have shown that when VDAC1 expression is increased, oligomerization forms large pores, opening MCU (mitochondrial calcium uniporter) and coupling MUC to IP_3_Rs, which leads to mitochondrial calcium uptake and mitochondrial calcium overload. It can promote the oligomerization of Bax/Bak to form a pore or open the mitochondrial permeability transition pore or Mitochondrial permeability transition pore (mPTP) [60]. Bax and Bak are two key pro-apoptotic proteins of the Bcl2 family. In healthy cells, Bak is distributed in the outer mitochondrial membrane, and Bax is distributed in the cytoplasm. After Cyt C enters the cytoplasm, it binds to Apaf1 and activates Caspase-9, which, in turn, activates Caspase-3, thus initiating the apoptotic cascade and causing apoptosis [61].

Interestingly, excess calcium levels trigger mitochondrial stress, mitochondrial dysfunction, and even neuronal death. Low mitochondrial calcium levels can also affect the viability of nerve cells. For example, in Parkinson’s disease, *α*-synapsin has been confirmed to be localized to MAM [62,63]. Studies have shown that the overexpression of α-synapsin in MAM will destroy the VAPB (ER protein capsule) [64]. VAPB (vesicle-associated membrane protein-associated protein B)—PTPIP51 (mitochondrial outer membrane protein, protein tyrosine phosphatase interacting protein 51) tethers affected ER-mitochondrial association, thereby affecting calcium signaling between mitochondria and endoplasmic reticulum and ATP production [65,66,67]. Studies have shown that no matter what kind of *α*-synapsin mutation, such as familial mutation or wild-type changes, can affect the VAPB–PTPIP51 tether; the result is the amount of calcium delivered by the ER to the mitochondria, which is reduced, and thus the calcium conduction is delayed, and because the mitochondrial tricarboxylic acid cycle is also regulated by Ca^2+^, resulting in reduced mitochondrial ATP production, which poses a serious threat to the viability of neurons [68,69].

Given the ubiquity of Ca^2+^ signaling in biology and its involvement in the etiology of PD and other neurodegenerative diseases, a growing body of theory now supports the idea that Ca^2+^ imbalance may be a key factor in aging. As discussed in this review, Ca^2+^ homeostasis regulation involves plasma membrane and intracellular calcium storage and the complex coordination within organelle networks; ultimately, the realization of metabolic interactions, intracellular signaling, cell maintenance, and cell survival regulation all require Ca^2+^ involvement. Understanding the mechanisms by which Ca^2+^ signaling promotes PD progression is therefore critical for the development of effective therapies for the disease.

## 3. Ferrum and PD

Iron is an indispensable trace element in cellular metabolism and is particularly important in the central nervous system [70]. Iron is involved in the development of brain synapses [71], myelination, and the generation and operation of neurotransmitters [72], which can affect the development of children’s IQ [73], cognition, movement, and social functions [74]. Studies have shown that ferritin levels in the brain increase with age and may be responsible for brain iron overload and neurodegenerative diseases in older adults [75,76]. Excess iron is linked to oxidative stress, inflammation, and cell death [77,78], and elevated iron levels have been detected in neurodegenerative conditions such as Parkinson’s disease and Alzheimer’s disease [79]. Cellular metabolism in the central nervous system requires iron as a redox metal to generate energy, primarily ATP, but neural tissue is susceptible to oxidative damage due to excess iron deposition, ultimately resulting in fibrinolysis [80].

Interestingly, iron deposition in the central nervous system is not solely attributed to aging; it is also influenced by iron metabolism-related diseases such as HH (hereditary hemochromatosis) [81]. HH is an autosomal recessive disorder of iron metabolism from mutations in the HFE (homeostatic iron regulator) gene (most notably C282Y and H63D) that regulate hepcidin function [81]. Iron in the brain is majorly concentrated in the basal ganglia and SN. In systemic iron imbalance, the blood–brain barrier protects the brain from the effects. However, in the case of iron overload, the BBB can disrupt the facilitation of the excess entry of iron into the brain. Patients with HH may be at an increased risk of PD due to their dysregulation of iron storage and metabolism [82].

Ferroptosis is an emerging caspase-independent RCD (regulated cell death) form proposed by Dr. Brent R. Stockwell, which is an iron-dependent form of cell death distinct from apoptosis, cell necrosis, and autophagy. There is no significant plasma membrane rupture and bioenergetic depletion in ferroptosis, and unlike other cell death morphologies, ferroptosis has smaller mitochondria than normal mitochondria and is characterized by reduced mitochondrial density [83], reduced or missing mitochondrial crest, and rupture of the outer mitochondrial membrane. In neurodegenerative diseases, ferroptosis triggers a series of events, including inflammatory activation, neurotransmitter oxidation, neuronal communication failure, myelin degeneration, astrocyte dysregulation, and nerve cell death [84,85]. 

It has been shown that accumulation of glutamate, iron, and polyunsaturated fatty acids, or depletion of GSH (Glutathione), NAD(P)H (Nicotinamide adenine dinucleotide phosphate), and GPX4 (Glutathione peroxidase 4) can trigger ferroptosis [86]. Iron homeostasis regulation is particularly important in the process of ferroptosis, and mutations in genes involved in iron metabolism may lead to increased intracellular iron input or decreased iron output [87]. The import of iron begins with the binding of TFR1 (Transferrin receptor 1) and subsequent endocytosis in endosomes. In acidic endosomes, Fe^3+^ is reduced to Fe^2+^ by STEAP3 (six-transmembrane epithelial antigen of prostate 3) and transported to the cytoplasm by DMT1 (divalent metal transporter 1) [88]. Iron in the form of Fe^2+^ can then be stored in ferritin or retained in the cytoplasm as a LIP (labile iron pool). Excess intracellular Fe^2+^ can directly catalyze the generation of lipid-free radicals through the Fenton reaction. PUFA (polyunsaturated fatty acid) is the main substrate for lipid peroxidation, and the plasma membrane contains a large amount of PUFAs and lipid-free radicals. PUFAs undergo peroxidation with lipid radicals to form PL-OOH (lipid peroxides). PL-OOH then cleaves, producing highly electrophilic secondary oxidation products [89], including epoxy, oxo, or aldehyde groups, which are highly reactive and toxic to membranes and cells. Eventually, nanometer-sized pores form on the cell-surface membrane, mediating the cell break-up that triggers ferroptosis [90]. Peroxidation-induced signals propagate from cell to cell in a unique wave-like diffusion pattern before cell breakdown as iron and lipid peroxidation persist, repeatedly creating a vicious cycle leading to cell death [91].

At present, there are three ways to prevent ferroptosis: the GSH/GPX4 pathway, the recently discovered FSP1/CoQ10/NAD(P)H pathway, and the GCH1 (GTP cyclohydrolase 1)/BH4 (Tetrahydrobiopterin)/DHFR (Dihydrofolate reductase) pathway (Figure 2) [92].

GPX4 (glutathione peroxidase 4), the star molecule for the study of ferroptosis, is an indicator of ferroptosis. Under physiological conditions, GPX4 oxidizes glutathione and reduces lipid peroxide (PL-OOH) associated with iron death to equivalent alcohol (PL-OH); thus, GPX4 activity is crucial in ferroptosis [93]. In GPX4’s fight against ferroptosis, glutathione is an essential substrate of GPX4. GSH is a linear tripeptide composed of glutamic acid, cysteine, and glycine, of which the thiol group on cysteine is considered its active site [94]. Its synthesis is influenced by the system XC-(cystine/glutamateantiporter), which regulates intracellular cystine levels [95]. Cystine enters the cell via system XC- and is then reduced to cysteine under the action of GSH or TRR1 (the enzyme thioredoxin reductase 1) [96]. Then, in the presence of ATP (adenosine triphosphate), GCL (glutamate cysteine ligase) catalyzes the combination of glutamate and cysteine to form γ-glutamylcysteine molecules. Finally, γ-glutamylcysteine is catalyzed by GS (glutathione synthetase) to connect with glycine to form GSH. In the biosynthesis of GSH, several transcriptases play a great role, such as GCL and GS. The synthesis of these enzymes is related to Nrf2 (nuclear factor-E2-related factor 2), and Nrf2 also exhibits strong antioxidant activity, which mainly relies on the expression of downstream gene proteins such as HO-1 (Heme Oxygenase-1), NQO1 (NAD(P)H quinone dehydrogenase 1) [97,98], GSTs (Glutathione-S-transferses), and GCLC (Glutamate cysteine ligase catalytic). Modulating Nrf2 expression may offer a new therapeutic approach to neurodegenerative diseases such as PD.

The FSP1-CoQ10-NAD(P)H pathway exists as an independent parallel system that cooperates with GPX4 and glutathione to inhibit phospholipid peroxidation and ferroptosis [99]. FSP1 (ferroptosis suppressor protein 1), discovered to be a glutathione-independent ferroptosis suppressor, contains an N-myristoylation signal and a flavoprotein oxidoreductase domain, which are thought to be critical sites for the inhibition of ferroptosis. CoQ10 (Coenzyme Q10) has long been shown to be a mobile lipophilic electron carrier, an endogenous synthetic fat-soluble antioxidant that plays a critical role in the mitochondrial electron transport chain [100,101]. Recent studies have demonstrated that CoQ10 can also act as a lipophilic radical scavenger antioxidant in plasma membranes [102,103]. FSP1 can prevent lipid oxidation by reducing CoQ10 with the assistance of NAD(P)H, and CoQ10 and its reduced form CoQ10-H2 can scavenge free radicals on the membrane surface and protect the integrity of lipoprotein and phospholipid molecules on the cell membrane [104]. In addition, NQO1 is a well-known NAD(P)H-dependent oxidoreductase that plays a role in ferroptosis protection by acting synergistically with FSP1 and regulating the reduction of ubiquinone to ubiquinol [105]. Interestingly, Nrf2 is also the central control factor for NQO1 expression [106]. The FSP1-CoQ10-NAD(P)H pathway still needs to be studied as a treatment for ferroptosis, but the FSP1-CoQ10-NAD(P)H pathway is second only to the GPX4/GSH pathway in the prevention of iron death to some extent and has great potential.

At present, there is minor research on combating ferroptosis through the GCH1/BH4/DHFR pathway in neurodegenerative diseases [107]. BH4 is a central factor in this pathway, playing a key role in biological processes and pathological states associated with the formation of certain neurotransmitters, immune responses, and pain sensitivity, and is an important cofactor in dopamine synthesis. BH4 is an antioxidant that traps lipid peroxidation radicals, while also promoting CoQ10 synthesis by converting phenylalanine to tyrosine, another way it exerts antioxidant effects. GCH1 is a rate-limiting enzyme for the synthesis of BH4, and the recovery and reuse of BH4 requires the participation of DHFR [108]. So, the DHFR inhibition can work together with GPX4 inhibitors to induce ferroptosis [109]. At present, the potential physiological and pathological role of GCH1/BH4/DHFR system in iron death and its application in Parkinson’s disease remain debatable.

In view of the mechanism of ferroptosis, we summarized the main therapeutic approaches for ferroptosis. We suggest that iron chelators, antioxidants, GPX4 activators, and NAD(P)H therapeutic agents may be more effective. We have collected the drugs targeting ferroptosis in PD in the past five years, and the specific content is in Table 1.

## 4. Calcium and Iron Crosstalk

The aforementioned statement highlights the role of iron in facilitating the generation of reactive oxygen species (ROS), which play a crucial physiological function in neuronal cells. ROS participates in the functional and structural modifications required for synaptic plasticity. Moreover, ROS levels regulate the function of the neuronal NMDA-R, Ca^2+^ channels, K^+^ channels, and CaMKII (calcium calmodulin dependent protein kinase II), among other cellular signaling enzymes. In particular, ROS can facilitate redox modification of RyR in hippocampal neurons, leading to activation of RyR-mediated Ca^2+^ release, thereby inducing phosphorylation of two enzymes involved in synaptic plasticity. Similarly, iron-induced ROS production can also trigger RyR-mediated Ca^2+^ signaling that promotes ERK1/2 phosphorylation in primary hippocampal cultures under Ca^2+^-free conditions. 

However, it is indisputable that high levels of ROS exert deleterious effects on neuronal cells. If the iron-induced release of Ca^2+^ mediated by RyR is enhanced, subsequent mitochondrial fission may also contribute to neuronal dysfunction. Excessive iron-induced peroxidation damages mitochondria, resulting in elevated levels of mitochondrial Ca^2+^, which subsequently stimulate the Ca^2+^-dependent phosphatase calcineurin and leads to neuronal cell death. It should be noted that the interaction between iron and Ca^2+^ signaling is bidirectional, as numerous proteins involved in cellular antioxidant defense and ROS production are dependent on Ca^2+^. In addition, lipid peroxidation products such as 4-HNE stimulate the opening of plasma membrane Ca^2+^ channels, including VGCCs in hippocampal neurons. Additionally, elevated 4-HNE levels can provoke energy dyshomeostasis and neuronal death by diminishing the functionality of Na^+^- and Ca^2+^-pumps and by altering Ca^2+^ channel permeability. It is important to note that the crosstalk between iron and Ca^2+^ signaling is bidirectional, as many proteins involved in cellular antioxidant defense and ROS production are Ca^2+^-dependent. An abnormal increase in intracellular calcium or an increased influx of calcium within mitochondria leads to more ROS production, which can lead to a disorder of the iron pool. Moreover, an increase of Ca^2+^ in the cytoplasm stimulates the neuronal nitric oxide synthase and the NAD(P)H oxidase, two enzymes involved in ROS generation and synaptic plasticity.

The above description points to a significant crosstalk between Ca^2+^ and iron. Thus, when one of the components is dysregulated, the other component is also affected, resulting in neuronal death.

## 5. Conclusions and Future Perspectives

PD is a common neurodegenerative disease, the true cause of which remains unknown. Metal elements’ ions may play an essential and irreplaceable role in the occurrence and development of Parkinson’s disease, of which the role of calcium and iron is particularly prominent. As we describe in this review, calcium, regulated by the plasma membrane and intracellular stored organelles (endoplasmic reticulum, mitochondria, etc.) is an indispensable signaling molecule in biological activity. Ca^2+^ homeostasis regulation is crucial for nerve cell activity, and the anti-neurodegenerative effects of calcium-channel blockers have brought light to us that microscopic exploration of Ca^2+^ signaling is crucial for the development of effective therapies for PD. Parkinson’s disease patients have a large amount of iron deposits in the substantia nigra, and ferroptosis is also an important factor influencing the occurrence and development of Parkinson’s disease. In the study of ferroptosis, most studies focus on the GPX/GSH pathway, and most of the studies in the past five years are Nrf2 agonists. While this is a truly effective and classical pathway, the therapeutic potential of the FSP1/CoQ10/NAD (P) H pathway is also of interest, but the key question here should be how to address the problem of CoQ10 synthesis at the cell membrane. BH4 has been described in dopamine synthesis and may also be valuable for the study of the significance of ferroptosis as well as the effect of PD. In our review, we have identified a crosstalk between calcium and iron signaling pathways, indicating that the imbalance of one pathway can significantly impact the balance of the other. Therefore, it is imperative to develop multifunctional drugs that target both Ca^2+^- and iron-dependent mechanisms.

## Figures and Tables

**Figure 1 brainsci-14-00088-f001:**
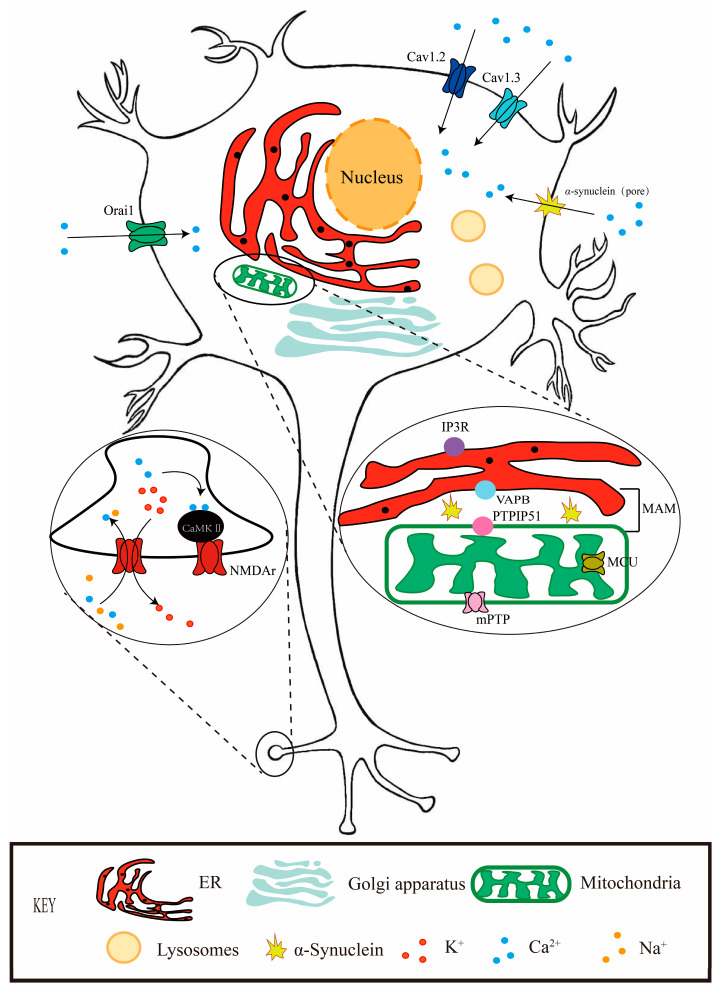
Ca^2+^ signaling and homeostasis in neuron. Schematic diagram of a neuron illustrating several related proteins and pathways affecting Ca^2+^. Ca^2+^ enters cells via Cav1.2, Cav1.3, α-synuclein-permeable pore, NMDA-R, and Orai1. The endoplasmic reticulum is the main intracellular Ca^2+^ reservoir, and its calcium regulation is affected by Orai1 and STIM. Calcium exchange between endoplasmic reticulum and mitochondria is mainly through MAM. Calcium ions in the ER are released directly by IP3R from the ER into the cytoplasm, and then into the mitochondria via VDAC1, a calcium channel on the outer mitochondrial membrane. Calcium overload caused by VDAC1 oligomerization and MCU opening opens mPTP and leads to cell apoptosis.

**Figure 2 brainsci-14-00088-f002:**
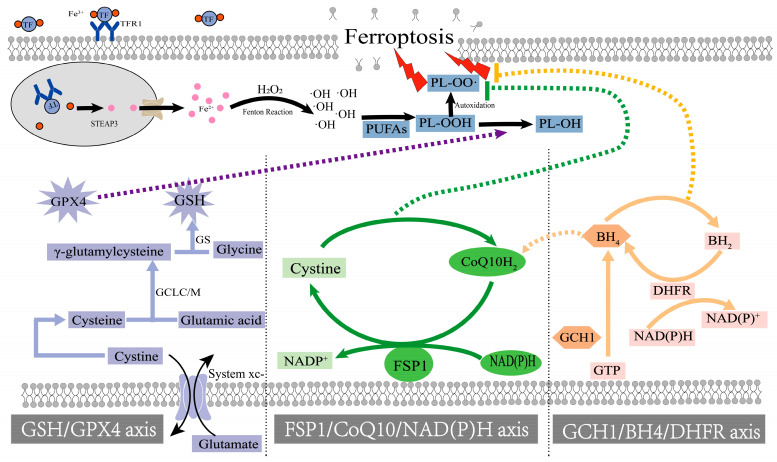
Ferroptosis-suppressing pathways. The input of iron begins with TFR1, and Fe^3+^ is reduced to Fe^2+^ by STEAP3. Excess Fe^2+^ directly catalyze PUFA to generate lipid-free radicals through Fenton reaction, resulting in cell membrane destruction. Glutathione synthesis begins when cystine enters the cell via system XC- and is then synthesized in the presence of GSH, TRR1, GCL, and GS. GPX4 oxidizes glutathione and reduces PL-OOH associated with iron death to an PL-OH. FSP1 reduces CoQ10 under the action of NAD(P)H. CoQ10 and its reduced form Coq10-H2 remove free radicals on the surface of the membrane. BH4 traps lipid peroxidation free radicals and promotes the synthesis of CoQ10. The synthesis of BH4 requires the participation of GCH1, while the recovery and reuse of BH4 requires the participation of DHFR.

**Table 1 brainsci-14-00088-t001:** Medicine targeting ferroptosis and its mechanism in the last five years.

Medicine	Mechanism and Function
Alpha lipoic acid	Antioxidant and iron chelator; regulates iron metabolism and mitigating ferroptosis through the SIRT1/Nrf2 signaling pathway [110].
Buddlejasaponin Ivb	Suppressed IRP2 (iron responsive element binding protein 2)-mediated iron overload [111].
Deferiprone	Iron chelator; inhibits pathological toxicity of α-syn in a mouse model of sporadic PD [112].
Desferrioxamine	Iron chelator; chelates irons [113].
Dl-3-n-butylphthalide	Regulates FTH (ferritin) expression, promotes Nrf2 nuclear translocation, and inhibits NCOA4-mediated ferritinophagy [114].
Doxycycline and Demeclocycline	Prevent intracellular oxidative stress and mitochondrial membrane depolarization [115].
Gastrodin	Antioxidant; increases the protein expression of Nrf2, GPX4, ferroportin-1 (FPN1), and HO-1 [116].
Hinokitiol	Antioxidant and iron chelator; chelates irons and activates cytoprotective transcription factor Nrf2 [117].
Icariside II	Antioxidant; activates Keap1/Nrf2/GPX4 signaling [118].
Idebenone	Inhibits the decrease of expression of NAD(P)H dehydrogenase, decreases the levels of the lipid peroxidation products, and increases the expression of GPx-4 [101].
Lapatinib	Activates GPX4/GSH/NRF2 axis; inhibits oxidative markers, including iron, TfR1, PTGS2, and 4-HNE; and suppresses p-EGFR/c-SRC/PKCβII/PLC-γ/ACSL-4 pathway [119].
Morroniside	Antioxidant; activates the Nrf2/ARE signaling pathway to protect dopaminergic neurons from ferroptosis in PD [120].
Paeoniflorin	Antioxidant; activates the Akt/Nrf2/Gpx4 pathway [121].
Pazopanib	Targets HSP90/CDC37 and its multiple RCD mechanisms [122].
Probiotic Strain L. lactis MG1363-pMG36e-GLP-1	Antioxidants and FSP1; activate the Keap1/Nrf2/GPX4 signaling pathway to down-regulate ACSL4 and up-regulate FSP1 to suppress ferroptosis [123].
Quercetin	Antioxidant; inhibits ferroptosis by activating the Nrf2 protein [124].
Rapamycin	Autophagy inducer; inhibits ferroptosis by activating autophagy [125].
β-hydroxybutyrate	Alleviates oxidative stress and ferroptosis via modulating ZFP36/ACSL4 axis [126].

## Data Availability

No new data were created or analyzed in this study. Data sharing is not applicable to this article.

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
