# Peer review of "The Role of Calcium and Iron Homeostasis in Parkinson’s Disease"

_brainsci, 2024, doi:10.3390/brainsci14010088_

Round 1
Reviewer 1 Report
Comments and Suggestions for Authors
The manuscript entitled "The role of calcium and iron homeostasis in Parkinson's disease" by Wang et al is a narrative review, exploring the involvement of neuronal calcium signalling and of iron regulation in the pathology of Parkinson's disease (PD). This review is well-organised and well-presented, incorporating two informative figures. Apart from some key information that is not included, the literature reviewed is comprehensive and unbiased. This work is likely to be of interest to readers working in the areas of neurodegeneration, neurobiology, cell signalling, cell death and iron homeostasis. However, a key omission is that relationships between calcium and iron homeostasis, and the role of these in in the development of PD, are not overtly stated. Since this would represent a major novel aspect of the review, this topic should be addressed.
MAJOR POINTS
1) Multiple publications in the field of ferroptosis indicate an key role of aberrant Ca2+ signalling in iron-dependent cell-death. In the current review, the relationships between calcium and iron homeostasis, and their involvement in PD, should be clearly described. Without making this link, the review appears to describe the effects of two unrelated ions in this neurodegenerative disease: this is clearly not the case.
2) If iron deposition promotes PD, are there associations between inherited disorders of iron homeostasis (such as haemochromatosis) and PD? If so, this aspect should be covered more comprehensively.
3) The review describes the release of cytochrome C (Cyt C) during mitochondrial Ca2+-overload-dependent apoptosis. It is important to point out that Cyt C promotes Ca2+-release via IP3Rs, thereby contributing to a positive feedback in this cell-death process.
4) Another interaction between apoptosis and Ca2+ signalling is that caspase-3 cleaves IP3Rs. However, the role of this proteolysis in promoting further apoptosis is controversial.
MINOR POINTS
1) Throughout, the 3 of IP3 should be subscripted.
2) Throughout, there is excessive use of the term "etc.". It is much better to list a few of the examples of what is being described, rather than using a vague term like etcetera (or similar terms like "various diseases" (line 117)).
3) Line 205, change "density concentration" to "density" (as density and concentration are synonymous in this context).
4) Line 225, change "cytoplasmic membrane" to "cell-surface membrane".
Comments on the Quality of English LanguageThe quality of scientific English is good. Only minor editing is required.
Reviewer 2 Report
Comments and Suggestions for Authors
1. The article covers a wide range of topics, from calcium signaling and iron homeostasis to ferroptosis and neurodegeneration. However, the transitions between sections and topics could be smoother to improve readability and flow.
2. The use of figures and diagrams is good, but additional graphical representations, especially ones that clearly depict the mechanisms discussed, could aid in understanding complex processes.
3. Elaborate further on the mechanisms by which Ca2+ influences neurotransmitter release, emphasizing its role in dopaminergic neuron activity and how disruptions may contribute to Parkinson’s disease. Suggested articles: https://doi.org/10.22088%2Fcjim.11.1.28
4. Provide more specific details about the involvement of different Ca2+ channels and pathways in Parkinson’s disease. This specificity can enhance the depth of understanding for readers with varying levels of expertise.
5. Clarify how the potential therapeutic pathways, particularly the FSP1/CoQ10/NAD(P)H pathway, could be practically applied or targeted in Parkinson’s disease treatment.
6. An additional reference that could make the ferrum and PD section more meaningful is this recently published review article. https://doi.org/10.1007/s11064-023-04032-5 This article could make your manuscript more meaningful as it provides a variety of essential oils that have anti-Parkinson’s effects. I suggest using this site in the last paragraph in this section, since you mention antioxidant effects. Additional articles that also discuss the antioxidant potential and activity of certain plants includes https://doi.org/10.59049/2790-0231.1164
7. The manuscript should expound further on the concept of autonomous pacing, providing a comprehensive elucidation encompassing both its general and specific definitions within the given context, in line 87.
8. The manuscript underscores the significance of dihydrofolate reductase (DHFR) in the process of tetrahydrobiopterin (BH4) reuse. Nevertheless, to enhance comprehension, it is imperative to integrate further details about DHFR, along with relevant information concerning GCH1.
9. Throughout the manuscript, there’s an inconsistency in the use of introduced acronyms, which disrupts reading flow. For example, “PD” and “Parkinson’s Disease” are used interchangeably. This also occurs in other terms as well such as “ER” and “Endoplasmic Reticulum”.
10. In line 158, precisely elucidate the identity of Bax/Bk.
11. In Figure 1’s legend it is mentioned that mitochondrial calcium homeostasis is affected by the “MUC” channel. Throughout the manuscript the acronym MCU was used. Please use the same acronym throughout the manuscript because it can confuse readers.
12. Line 69, ER acronym has not been introduced prior.
13. Line 124, AD acronym has not been introduced prior.
14. Line 144, there is no need to reintroduce what MAM acronym means.
Reviewer 3 Report
Comments and Suggestions for Authors
The manuscript by Ji Wang et. al., titled “The role of calcium and iron homeostasis in Parkinson's disease” is a systematic review of the current status of calcium and iron in Parkinson disease, can be considered for publication with some major edits.
Major comment:
1. Criteria for selecting articles for the review should be mentioned
2. A tabular column can be made summarizing all the research articles mentioned on roles of Calcium and Ferrum on PD, with kind of model used and other useful information which will help in better understanding of the work done till date.
3. Some of the recent article should be included and discussed
a. Ferroptosis as a New Mechanism in Parkinson’s Disease Therapy Using Traditional Chinese Medicine, by Lijuan Wu, 2021;
b. Ferroptosis in Parkinson & Disease: Molecular Mechanisms and Therapeutic Potential’ by Ding XS, 2023
c. Parkinson’s disease, epilepsy, and amyotrophic lateral sclerosis—emerging role of AMPA and kainate subtypes of ionotropic glutamate receptors by Marina N. Vukolova,2023.
4. Multiple grammatical mistakes found throughout the manuscript. The author should rephrase multiple sentences instead of repeating the similar adjectives in every paragraph.
Minor comment:
1. In Lines 54 and 55, “In neurons, Ca2+ motion can occur across the plasma
2. membrane, either through electrical activity or through agonists” the authors need to specify which agonists.
4. The author needs to clarify whether MCU or MUC. In Figure 1 it is MUC line 69 it is MCU.
5. The authors need to correct the starting and closing of the brackets and punctuation in a meaningful way in Lines 65-67. Please check is it transporter or receptor
6. The author needs to maintain uniformity in using the terms. For example, Figure 1 legend ‘NMDAr’ but in line 67 it is NMDA-R. Similarly, Orai1 in Figure 1 and ORAI1 in line 70.
7. Line 67- 68, please reframe the statement, extracellular to intracellular but mitochondrial is from intracellular
8. All Abbreviations must be expanded when used for the first time not anywhere else: SN, MCU, RyR and IP3, STIM/ORAI, etc GSH and GPX4 were used in line 211 but the abbreviation was explained in 243. Expnad VDAC1 in Line 147-159.GSH has not expanded at all. Similarly, BH4 and DHFR were used in 142 and Figure 2, but explained in Lines 147-159 only.
9. The authors used Cav1.2, and Cav1.3 for VGCC subunits. Refer to and correct suffix as per the new nomenclature.
10. Line 92-93 : please maintain the uniformity “Ca2+” or " Ca2+ "
11. ‘Ca2+ enter cells via Cav1.2, Cav1.3, α-synuclein-permeable pore, NMDAr, Orai1.’ This may be written as Ca2+ enter cells via Cav1.2, Cav1.3, α-synuclein-permeable pore, NMDAr, and Orai1’.
Round 2
Reviewer 1 Report
Comments and Suggestions for Authors
The revised manuscript is now suitable for publication in Brain Sciences
Comments on the Quality of English LanguageNA
Reviewer 2 Report
Comments and Suggestions for Authors
Thank you for taking all comments into consideration; the manuscript is better now.
Reviewer 3 Report
Comments and Suggestions for Authors
Authors have answered most of my comments satisfactorily and no more comments or suggestions.